# Robust motion detection and classification in real-life scenarios using motion vectors

**Sameed Ur Rehman**[1], **Irshad Ullah**[1], **Wajahat Akbar**[2], **Altaf Hussain**[2], **Tariq Hussain**[3], **Ahmad Ali Alzubi**[4], **Insaf Ullah**[5]*, **Shuguang Li**[2]

1 Institute of Management Sciences (IMSciences), Peshawar Pakistan, 2 School of Electronic and Control Engineering, Chang'an University, Xi'an, China, 3 School of Computer Science and Technology and School of Statistics and Mathematics, Zhejiang Gongshang University, Hangzhou, China, 4 Department of Computer Science and Engineering, College of Applied Studies, King Saud University, Riyadh, Saudi Arabia, 5 Insaf Ullah from Institute for Analytics and Data Science, University of Essex, Colchester, United Kingdom

* Insaf.ullah@essex.ac.uk

## Abstract

In dynamic settings such as security, autonomous driving, and robotics, effective motion detection and classification are crucial for accurate tracking amidst target and background movements. Traditional approaches, typically designed for static environments, face challenges in complex scenes with multiple types of motion. This research presents a robust algorithm for motion detection in fully dynamic scenarios, utilizing the macro block technique to generate motion vectors, followed by motion vector analysis to classify distinct types of motion. These include camera motion, object motion, background motion, and complex motion, where both background and foreground move simultaneously. By segmenting and categorizing these motion types, the proposed approach improves detection precision in cluttered, real-world environments. Furthermore, the algorithm adapts to lighting variations and is independent of specific sensor setups. Moreover, the high agreement with human judgment, achieving a 90% accuracy rate, underscores the model's robustness and potential applicability in real-world scenarios where dynamic backgrounds are prevalent. This establishes a framework for future research in dynamic motion detection and classification.

## 1 Introduction

The key technology that allows artificial systems to understand and interpret the visual environment is computer vision. Making well-informed judgments involves processing and analyzing images, videos, multicamera views, or multidimensional data such as medical images [1,2]. Motion detection, which recognizes moving objects in a scene and provides details of their shape, speed, and direction, is a crucial component of computer vision. Applications such as autonomous navigation, object tracking, and surveillance require this functionality [3]. Visual motion in practice can be

**Data availability statement:** The minimal dataset necessary to replicate the study findings is fully accessible to all researchers and can be accessed via the following link: https://zenodo.org/records/17217381.

**Funding:** This work was supported by Ongoing Research Funding program (ORF-2025-395), King Saud University, Riyadh, Saudi Arabia.

**Competing interests:** No competing interests.

caused by a variety of things, including changes in lighting, movements of the viewer, and movement of objects within the image. This intricacy makes the use of advanced techniques for precise object recognition and motion detection necessary [4]. In this work, we use dynamic videos that have different scenarios of motions and perspectives to identify motion. Ego-motion is ignored because the video formatting is standardized, but other elements like dynamic backgrounds and shifting lighting are taken into account. The main task of this research is to create a reliable system that can distinguish between the foreground and background in situations where both may be in motion. Conventional motion estimation methods often assume that while the foreground moves, the background stays unchanged [5,6]. However, this assumption falls short in scenarios where there is motion in the background, for example, in the rain or when the camera vibrates. Thus, the detection and categorization of complicated motion types in such hard contexts is the primary issue this work attempts to solve. We propose a sensor-free visual system that enables reliable motion detection for dynamic or targeted movements. To enable precise spatial motion analysis, we begin by acquiring frames from videos containing these motion types[7,8]. Each frame is processed individually, undergoing a standardization procedure to ensure uniform video formatting. Following standardization, we apply pixel intensity normalization to address lighting variability, providing consistent and normalized video frames for subsequent analysis[9]. This approach optimizes our ability to analyze and detect motion accurately in complex real-world conditions. A number of crucial problems are addressed by our suggested solution, including controlling non-uniform shifting backgrounds, identifying foreground and background based on motion and stillness, and allowing for progressive and worldwide changes in lighting. These developments offer a system that is highly adaptive to dynamic, unstructured settings, which makes a substantial contribution to the field of advanced computer vision. Extensive studies and applications using several test beds of virtual and real-life settings verify the effectiveness of our solutions. The main objectives of this study are:

- To develop a method to accurately identify rain or snow in the scene.
- To create algorithms to detect and compensate for camera motion in the scene.
- To separate different types of motion factors within the scene.
- To develop techniques to effectively isolate the background in various scenarios. The newness in this research stems from differentiating motions or identifying motion types in the video frames concerned. Here, we have identified four types of motion: object motion, background motion, camera motion, and combined motion (the motion of both foreground and background).

This new approach is a milestone as it helps us identify whether the object is moving within the defined frame, the background is moving, or the camera is being panned, because in interesting scenarios where both can be in motion, distinguishing between the camera and object motion becomes a tough task. The remainder of

this study is organized as follows: Sect 2 covers the related work, Sect 3 presents the methodology, Sect 4 details the experiment, results analysis, and discussion, and Sect 5 provides the conclusion.

## Contribution

The contribution of this work is to advance the field of motion detection and classification by addressing the challenge of identifying motion in fully dynamic backgrounds, i.e., scenarios in which both the foreground and background are moving. Unlike previous approaches, such as Gaussian Mixture Models (GMM) [10] and histogram-based methods that are primarily focused on distinguishing between foreground and background, this work presents a robust algorithm that is able to classify different types of motion, including object motion, background motion, and camera motion, in complex scenarios with multiple elements moving simultaneously. By using macro-block techniques to develop motion vectors and analyzing these vectors to classify the type of motion, the proposed approach brings significant improvements over existing methods. This innovation not only improves the accuracy of motion detection in complex and dynamic environments but also creates a comprehensive framework for understanding motion dynamics in real-world applications, paving the way for future advances in the field.

## 2 Related work

Motion detection research in computer vision has a long history, with early attempts focusing on methods like temporal differencing. One of the first techniques, temporal differencing, involves removing frames one after the other to emphasize changes and detect moving parts in video sequences. This method is very flexible in circumstances that change quickly. Nevertheless, this frequently results in the foreground aperture issue, as it is unable to fully catch the contour of some kinds of moving objects. Temporal differencing techniques typically utilize a threshold mechanism to detect motion by comparing changes between subsequent frames, and therefore require additional approaches to identify stopped sections [9,10]. A system inspired by Pfinder, a real-time system for tracking and analyzing human behavior, was created by Lipton et al. [11]. They used a combination of temporal and spatial differencing to track and categorize targets in video sequences. They successfully tracked classed targets throughout time and location by incorporating target classification metrics. Although robust, the system was less effective in highly dynamic scenarios due to its susceptibility to changes in appearance and background noise. Chang suggested a method for detecting changes that is just dependent on temporal variations [10]. This approach made it easier to partition moving objects into their foreground and background and allowed for continuous monitoring, which made it useful for video surveillance. But it had trouble tracking camera movement, and it needed a steady background to recognize objects correctly. Another well-liked method for motion detection is background removal. To detect moving objects, this approach compares the current frame with a background picture that serves as a reference. Even though they are simple to use, simple background subtraction techniques often perform poorly in real-world situations because of problems with dynamic backgrounds, changing lighting, and camera movement [12–14]. More complex models, such as the Gaussian Mixture Model (GMM), were created to solve these issues. GMM-based methods represent each pixel with several Gaussian distributions, providing resilience against noise, shadows, and illumination variations [9]. Kim expanded on the background subtraction method by including picture registration techniques and building panoramic background models. The goal of this technique was to address problems with background adaptation and camera movement [15]. Its high computational memory needs and startup delays remained a problem, though. Grimson [16] and other researchers enhanced background modelling even more by creating multi-color adaptive background models that could be tracked in real-time [17]. These models solve problems like as changes in background motion and lighting by updating the background for every frame in real time. These improvements did not eliminate the approaches' inability to cope with quick scene changes and high computational overhead. The capacity of dynamic threshold-based algorithms to identify moving objects in intricate environments with shifting illumination has drawn attention recently. With the use of these techniques, multi-target motion detection in dynamic backgrounds may be accomplished with more accuracy

by computing a dynamic threshold based on the gradient change between successive frames [17]. The literature demonstrates how motion detection approaches have developed over time, moving from basic temporal differencing to sophisticated background modelling techniques. Research in this field is continuing because, despite great advancements, difficulties are still encountered when effectively identifying and tracking moving objects in intricate and dynamic situations [18]. Table 1 shows the comparative analysis of motion detection algorithms used for motion detection and classification, and the proposed work.

## 3 Methodology

The aim of this study is to detect motion within a dynamic background, which refers to scenarios where both the background and the foreground are in motion. This investigation utilizes data from various sources, including the videos paris_cif.avi, Mixmasterfile.mpg, Aeroplanes.mp4, Wildlife.avi, and Pool.mp4.The study begins with data collection, specifically gathering the targeted videos. Once the videos are collected, preprocessing is conducted to refine the data for achieving the desired results. The first step in preprocessing involves extracting frames from the videos, followed by normalization of these frames. Normalization is a critical step in motion detection algorithms as it mitigates the impact of variations in lighting conditions, contrast, and exposure levels between frames. By normalizing pixel values, the algorithm can more effectively detect motion changes that are not merely due to brightness or contrast differences. Following normalization, the frames undergo RGB to grayscale conversion to standardize color intensity and eliminate color differences [27–30]. The subsequent step involves splitting the frames into macro blocks, which allows for determining a suitable macro block size for the algorithm. After defining the macro blocks, the motion vector for each block is estimated [31,32]. The

**Table 1**. Comparative analysis of motion detection algorithms and proposed work.

| References | Approach | Dataset | Research Aim | Motion Classification (Type) |
|---|---|---|---|---|
| [19] | ERD (ensemble random forest decision tree) | Accelerometer and gyroscope sensors data | Detect human kinematics motion such as walking or running with high performance | Walking and running of an object |
| [20] | YOLO, ResNet152 | CCTV traffic surveillance footage | Detect traffic incidents and classify them for an alarm system | Causes of accident |
| [21] | Niblack's threshold method | Surveillance videos | Handle illumination and background clutter challenges; classify motion/non-motion at runtime | Motion vs. non-motion |
| [22] | YOLOv8 agri + DeepSORT | Custom dataset | Address lighting effect limitations in object detection | Object detection considering lighting effects |
| [23] | Temporal difference | Static camera | Detect moving objects at distance for object tracking using frame differencing | Object detection using frame differencing |
| [24] | Infrared spectral imaging + LBP | Infrared spectral imaging | Improve target motion detection and segmentation using LBP features | Motion detection in infrared images |
| [25] | Three-frame difference algorithm | Custom dataset | Optimize video surveillance via motion detection for efficient storage/transmission | Motion detection for storage optimization |
| [26] | Frame difference + convolution | Jilin-1 | Improve moving vehicle detection in satellite videos by addressing blurred boundaries | Object detection in satellite videos with boundary refinement |
| **Proposed Methodology** | Macro-Block and Motion Vector Analysis | Mp4, Avi, MPG video formats | Detect and classify motion into various types | Background, foreground, complex motion |

 

motion vector indicates how pixels within a macro block move from one frame to another, enabling the video codec to encode the motion information rather than the actual pixel values, thereby enhancing compression efficiency. The final step is motion vector analysis, where the motion of each individual macro block across different frames is analyzed over a specified time frame and with the help of motion vector analysis we can differentiate the type of motion present in the specific scene in a video, all these steps are diagrammatically presented in the Fig 1.

### 3.1 Dataset descriptions

Five different video files are shown in detail in Table 2, together with information on each one's length, frame rate, number of frames, resolution, and pixel size per frame. With a total runtime of 1635.25 seconds, or around 27.25 minutes, the films provide a sizable dataset for motion detection research. The frame rates provide a diversity of motion smoothness for evaluating the algorithms' durability, ranging from 15 to 30 frames per second, with "Wildlife.avi" significantly slower at 29.97 frames per second. With "Mixmasterfile.mpg" giving the most frames (34,389) and "paris_cif.avi" the fewest (1,065), the overall amount of frames across all films is 41,110, demonstrating the variation in data volume. Three videos have a resolution of 1280x720 pixels, while the other two have a size of 352x288 pixels. This means that the material is a combination of high-definition and lower-quality. The disparity in quality and frame rate across the movies is essential for the creation and evaluation of computer vision models, which must function effectively in a variety of complicated and varied video scenarios. The name, length, frame rate, total number of frames, width, height, and pixel size of each video file are all specified in detail. This table effectively represents the general overview of the input videos used. We have five distinct video formats, including AVI, MP4, and MPG. There are 41,110 frames in all, and their combined length is 27.25 minutes, or 1635.25 seconds.

### 3.2 Preprocessing and feature extraction

The preprocessing and feature extraction steps are crucial for obtaining accurate results, as preprocessing prepares the data for analysis. In this process, the data is handled step by step, starting with frame extraction followed by converting RGB frames to grayscale, and then normalizing the frames to ensure consistency in the input data.

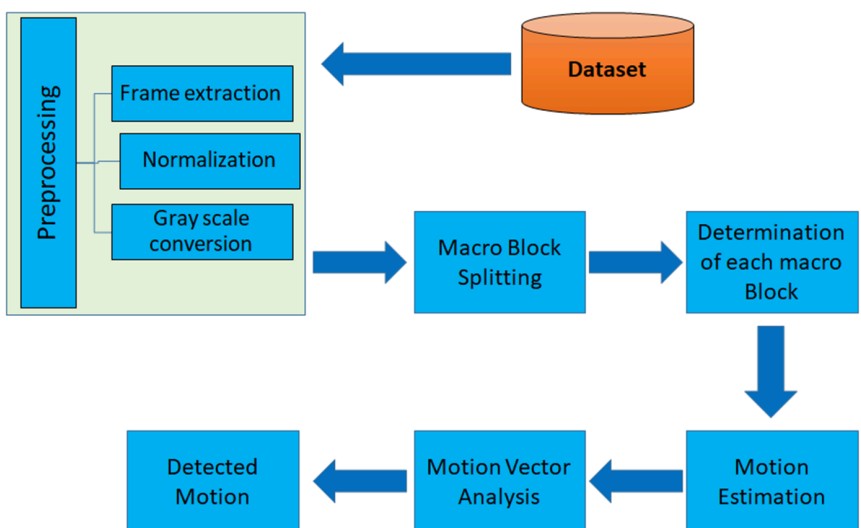

**Fig 1**. An overview of the proposed methodology, illustrating the workflow from data extraction to motion detection.

**Table 2**. Overview of video dataset properties, including duration, frame rate, total frames, resolution (width and height), and pixels per frame, summarizing five video files with a cumulative duration of 1635.25 seconds (27.25 minutes) and 41,110 frames.

| Video | Duration (s) | Frame Rate (fps) | No. of Frames | Width (px) | Height (px) | Pixels per Frame |
|---|---|---|---|---|---|---|
| paris_cif.avi | 71.00 | 15 | 1,065 | 352 | 288 | 101,376 |
| Mixmasterfile.mpg | 1375.66 | 25 | 34,389 | 352 | 288 | 101,376 |
| Aeroplans.mp4 | 68.31 | 30 | 2,049 | 1280 | 720 | 921,600 |
| Wildlife.avi | 30.10 | 29.97 | 902 | 1280 | 720 | 921,600 |
| Pool.mp4 | 90.19 | 30 | 2,705 | 1280 | 720 | 921,600 |
| **Total** | **1635.25 (27.25 min)** | – | **41,110** | – | – | – |

### 3.3 Frame extraction and normalization

For efficient motion detection in video processing, frame extraction and normalization are essential processes [33]. We take a sequence and use Fn to represent the n-th frame in order to retrieve frames. In practice, we compare frames separated by a gap D, such as F1 with F3 rather than consecutive frames, which are frequently excessively similar shown in Fig 2. This method improves the chances of finding notable motion variations.

To accommodate for changes in illumination and other environmental conditions, normalization adapts pixel intensity values to a standard scale. This preprocessing stage facilitates the algorithm's attention to real motion as opposed to variations in brightness. Frame differencing, mean normalization, and contrast stretching are a few examples of different

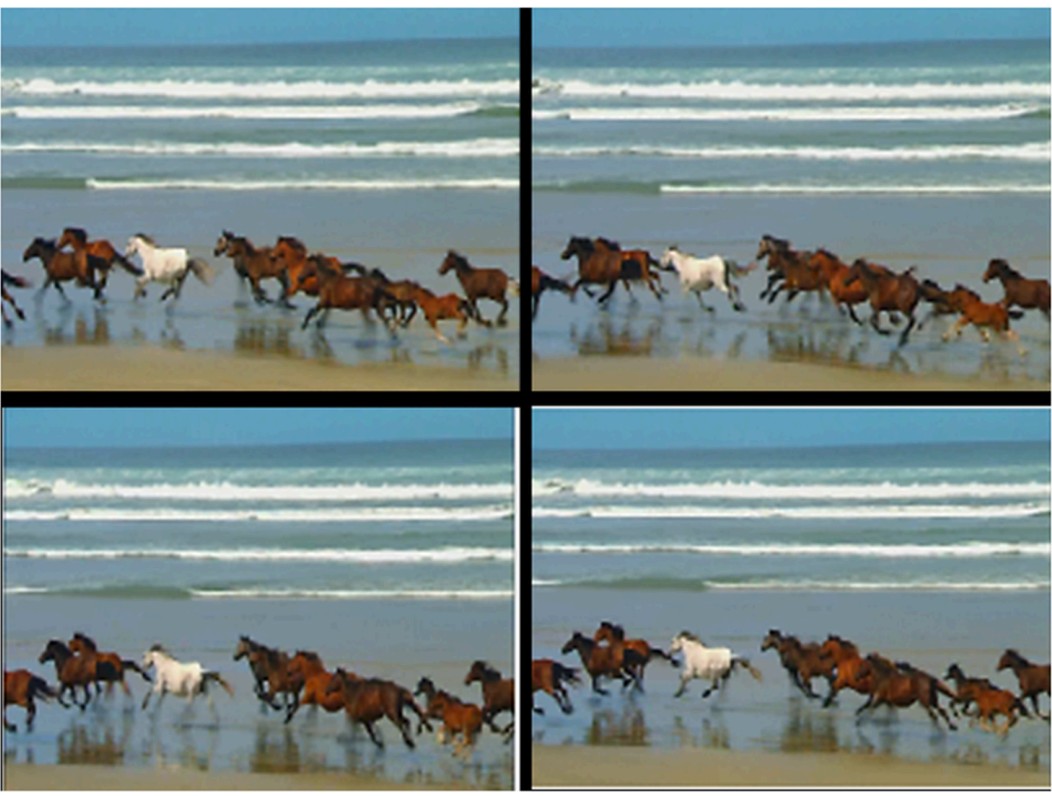

**Fig 2**. Extracting frames enables motion detection and classification by isolating key instances from a dynamic background, capturing temporal changes for accurate analysis.

normalization techniques. In order to emphasize changes, frame differencing entails subtracting the pixel values of succeeding frames Fig 2. In order to minimize the effects of illumination fluctuations and facilitate reliable motion detection across several video sequences, the normalization step is crucial.

### 3.4 Image resizing and grayscale conversion

Image Resizing and Grayscale Conversion We scale frames to a common 500×500×3 format so that films of varying resolutions, such VGA and HD, work together Fig 3. This resizing guarantees that every frame has the same size, which is important for further analysis and promotes consistent processing. We resize the frames and then transform them from RGB to grayscale Figs 4 and 5. By reducing the three color channels in the data to one intensity channel by grayscale conversion, problems with color disparities are avoided and intensity variations are the main emphasis, improving the accuracy of motion recognition.

### 3.5 Macro block size selection

Smaller macro blocks provide finer detail on motion but increase computation overhead. For example, a 50×50 frame contains 2,500 pixels; if it is divided into 5×5 blocks, it yields 100 macro blocks (2,500/25), thus allowing pixel-level analysis but takes longer to process. On the contrary, macro blocks measuring 25×25 will generate only 4 blocks (2,500/625), which in itself is too coarse to detect subtle motion variations. A compromise was therefore made to use a 10×10 macro block size. This means 25 blocks per 50×50 frame (2,500/100), which is a compromise between capturing detailed motion and efficiency in computation. Therefore, it is not too sensitive to noise arising from pixel-wise processing while still retaining enough detail to capture relevant motion patterns.

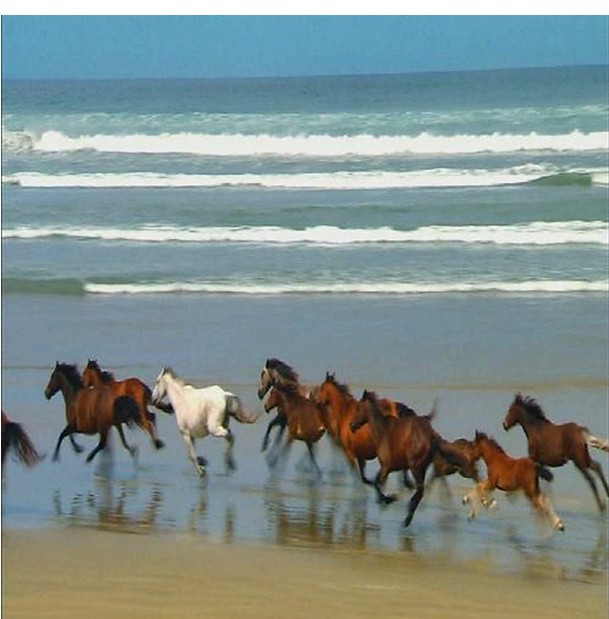

**Fig 3**. **Frame normalization to a uniform 500×500×3 resolution to ensure consistent motion detection and classification across varying video formats, such as VGA and HD, in dynamic backgrounds.**

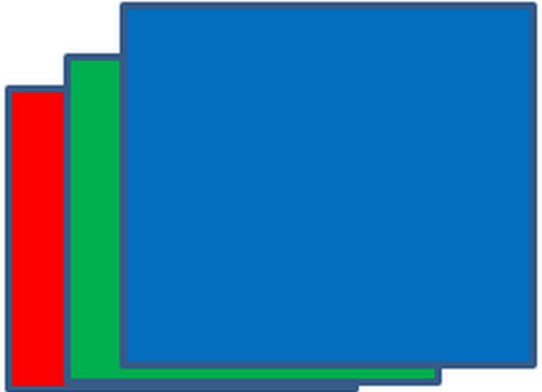

**Fig 4**. Transformation of a RGB image into a Grayscale representation.

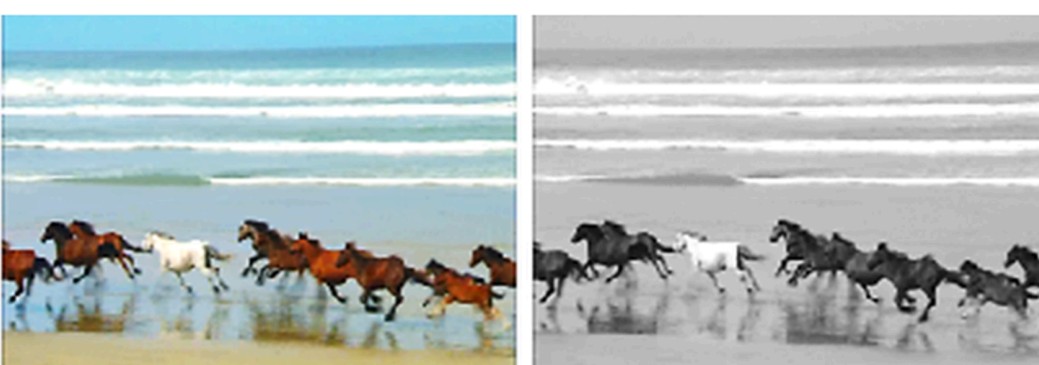

**Fig 5**. Transformation of an RGB image into a Grayscale Representation, (left side) shows the original image, (right side) shows the grayscale image.

### 3.6 Macro-block division and motion vector estimation

Macro-blocks are the basic building blocks of motion analysis, and they are separated into frames [34,35]. Since 10x10 is the preferred size for macro-blocks, there are 2,500 macro-blocks in each 50x50 frame Fig 6. This size strikes a compromise between the capacity to record minute motions and computational efficiency. We employ the differences between frames separated by a predetermined gap, d, for motion vector estimation. For best results, we set d to 5 frames. Polar coordinates are used to compute the motion vector shown in Eqs 1, 2, and 3 respectively, which indicates the displacement and direction of motion shown in Fig 7.

$$mv = (d, \theta) \qquad (1)$$

Where

$$d = \sqrt{(\Delta x)^2 + (\Delta y)^2} \qquad (2)$$

So, $\theta$ will be:

$$\theta = \tan^{-1}\left(\frac{\Delta y}{\Delta x}\right) \qquad (3)$$

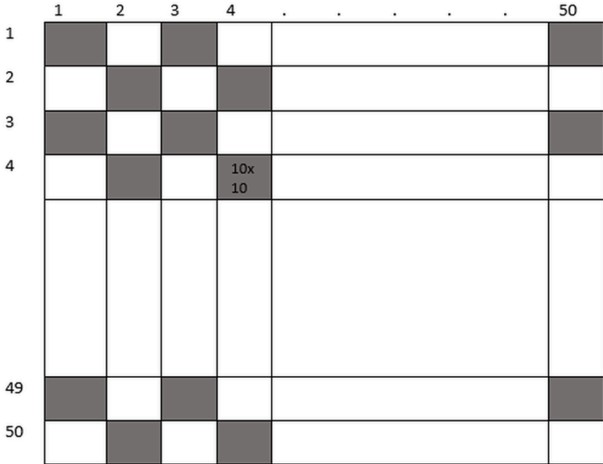

**Fig 6**. **50×50 Macro-Block representation in digital image processing.**

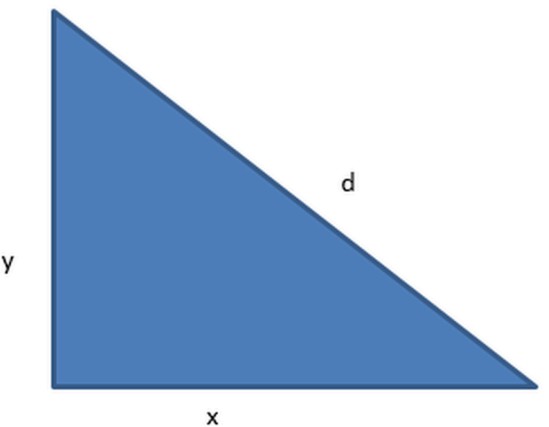

**Fig 7**. **Showing the tan$\theta$ in a diagrammatical way.**

### 3.7 Motion analysis

By comparing motion vectors between macro-blocks, motion vector analysis is able to distinguish between various motion situations [36–38]. For instance, when all macro-blocks have differences less than a threshold $d \leq th$, rather than being established as a constant, the threshold value (th) for each motion classification was dynamically defined. For each video, the mean ($\mu$) and standard deviation ($\sigma$) of motion vector magnitudes were computed, and the threshold was thus set by the formula: th = $\mu$ + 0.75$\sigma$. This data-driven strategy enables the algorithm to adapt to heterogeneous motion intensities and environmental noises, and consequently provides robust classification results. i.e.no motion Fig 8 is identified. When all macro-blocks exhibit considerable motion $di>th$, camera motion Fig 9 is indicated; when some blocks move but others stay static, object motion Fig 10 is noted. Accurate motion identification in complicated scenarios is made possible by the algorithm's ability to discern between moving objects, static backgrounds, and camera motion through the analysis of these patterns Fig 11.

**Case 1: No motion**

$$d(1 \ldots 2500) \leq th \tag{4}$$

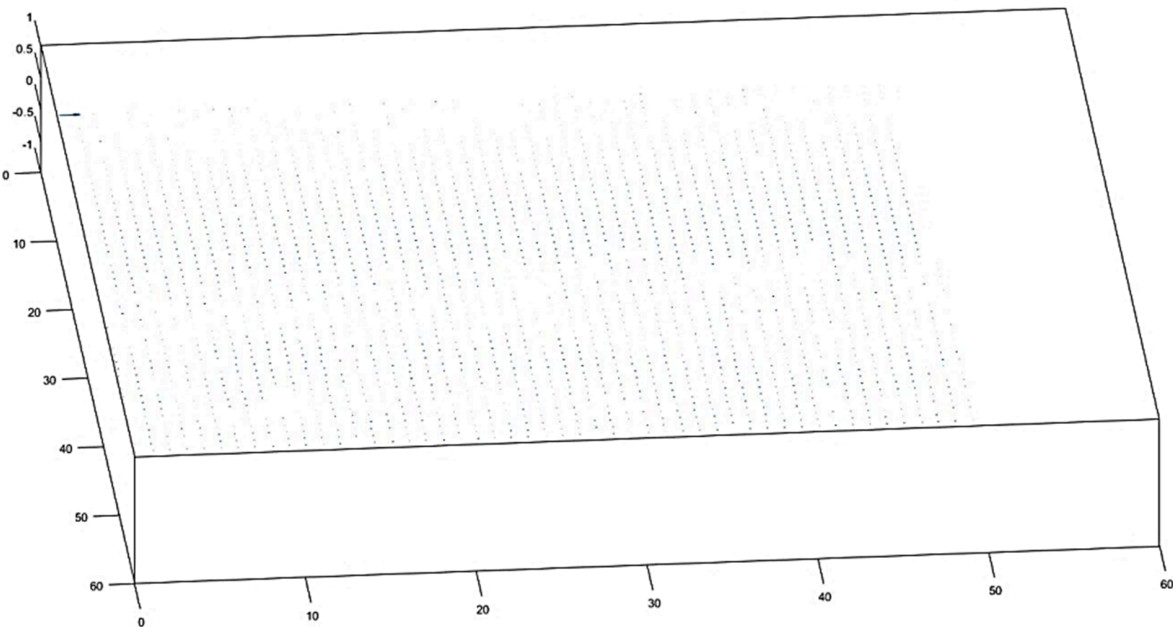

**Fig 8**. The scene remains completely still, with no motion detected.

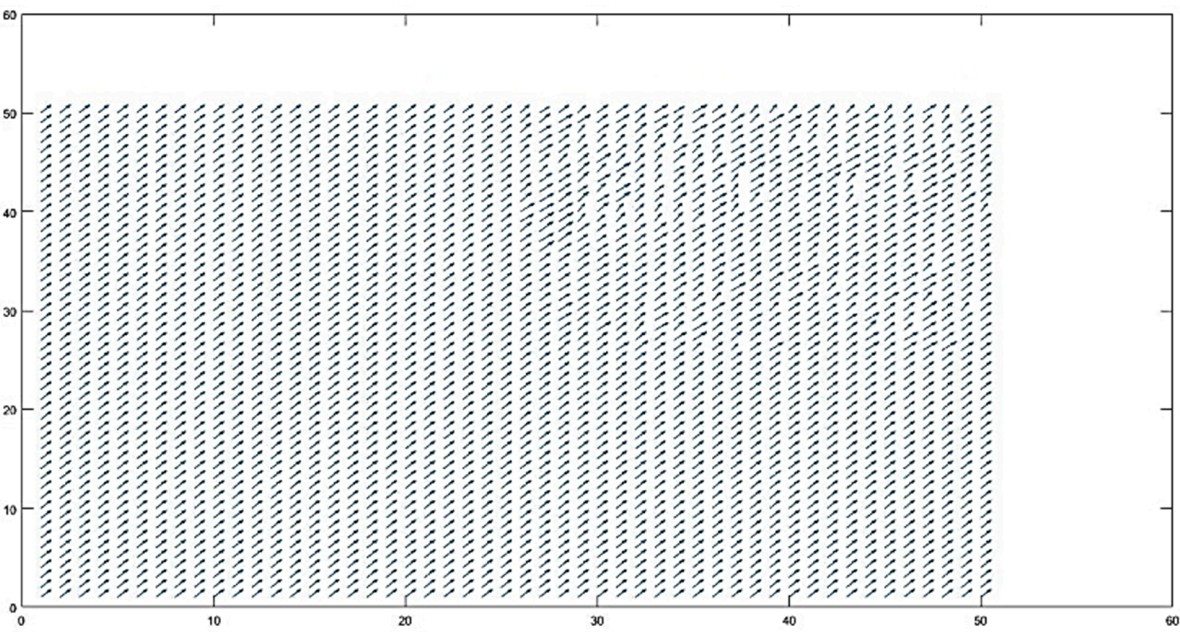

**Fig 9**. Scene remains static while the camera moves.

In Case 1, Eq 4 represents the analysis confirmed the complete absence of any detectable motion within the scene. All objects remained static, indicating a lack of activity or movement. This observation underscores the scene's stability and lack of dynamic elements shown in Fig 8.

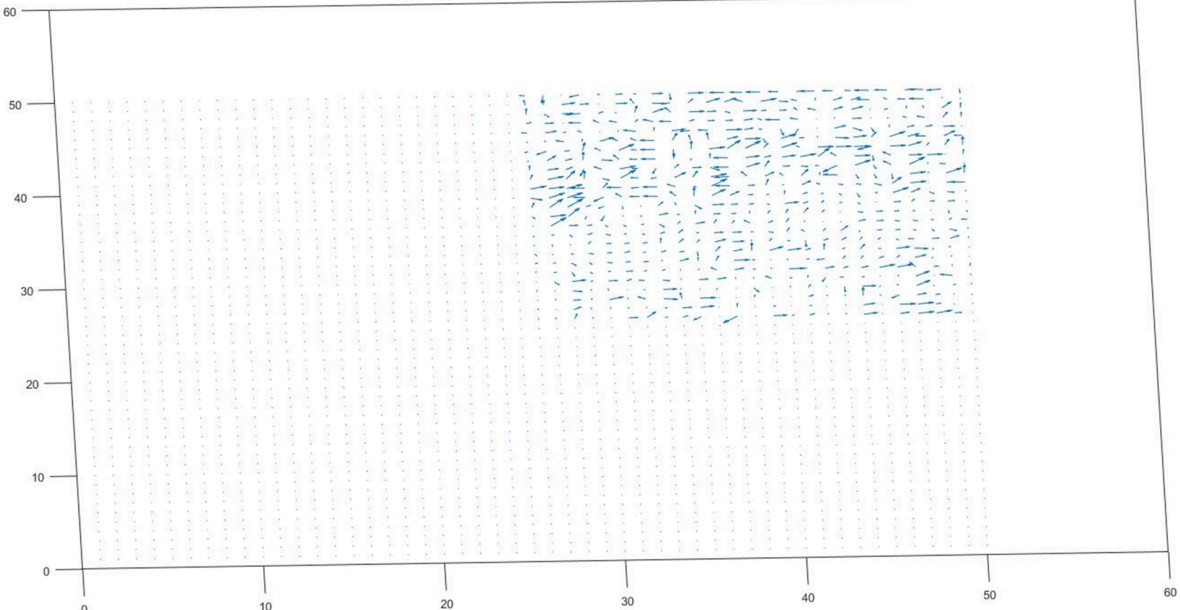

**Fig 10. Camera remains static while the object moves.**

**Case 2: Camera motion only**

$$d(1 \ldots 2500) > th \tag{5}$$

$$d_i \approx d_j \qquad \text{for } i \neq j \tag{6}$$

In Case 2, Eqs 5 and 6 respectively represents the analysis identified that the only detected movement was due to the camera itself. The objects within the scene remained stationary, while the camera's motion introduced the observed changes. This highlights that any dynamic elements are a result of the camera's activity rather than the scene's contents shown in Fig 9

**Case 3: Object motion**

$$\text{For some } d \leq th \tag{7}$$

$$\text{For some } d > th \tag{8}$$

In Case 3, considering Eq 7 is for background and Eq 8 is for object motion, the analysis revealed that an object within the scene was in motion. This indicates that there is active movement among the scene's elements, distinct from any camera movement. The presence of this moving object adds a dynamic aspect to the scene, differentiating it from static scenarios shown in Fig 10.

**Case 4: For all d**

$$d(1 \ldots 2500) > th \tag{9}$$

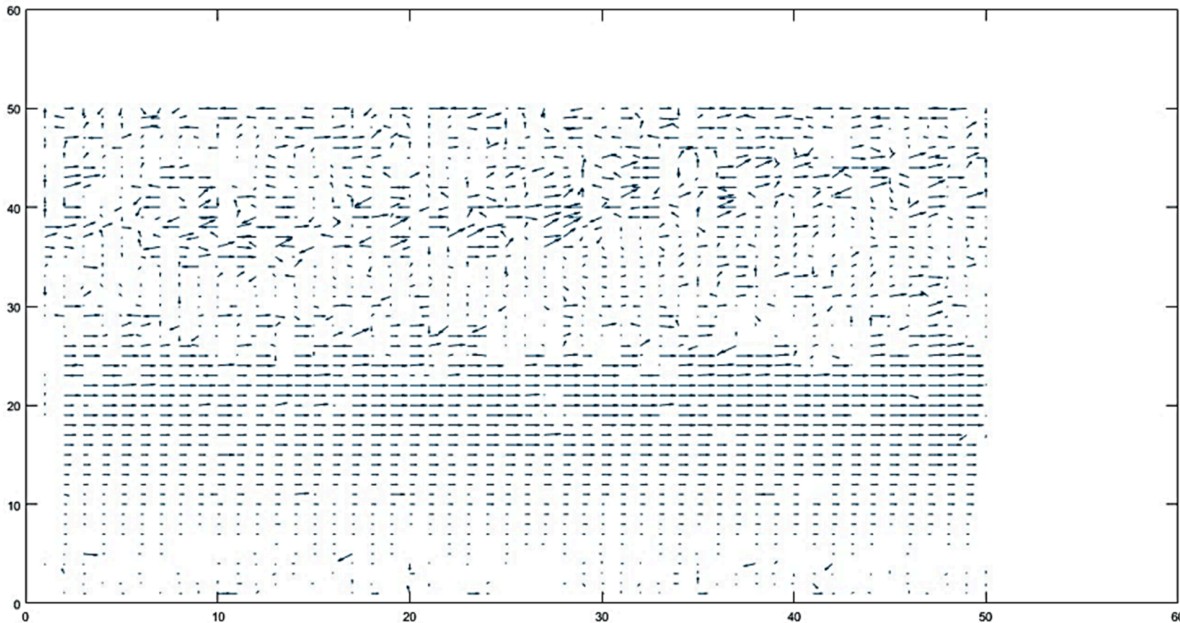

**Fig 11. Objects, background and camera all are in motion.**

In Case 4, considering Eq 9 where d_id_j the analysis showed simultaneous motion in the object, background, and camera within the scene. This combination of movements results in a highly dynamic and complex environment. The interplay of these elements creates a scenario where both the subject and surroundings are in motion, along with the camera capturing the scene shown in Fig 11.

### 3.8 Proposed model

The proposed methodology begins with extracting frames from a video in their original RGB format, which are then normalized to a fixed resolution of 500 × 500 pixels to ensure uniformity and efficiency. The RGB frames are subsequently converted into grayscale to reduce complexity while retaining essential structural details. Each grayscale frame is divided into grids of varying scales (5 × 5, 10 × 10, and 25 × 25). Among these options, the 10 × 10 grid is selected as the optimal choice. Choosing a 5 × 5 grid would result in a highly pixel-based analysis, leading to increased computational complexity. Conversely, selecting a 25 × 25 grid would likely overlook finer details and subtle motions, as the larger grid size might fail to capture small movements. Therefore, the 10 × 10 grid strikes a balance by effectively capturing minute motions while maintaining computational efficiency, enabling multi-resolution analysis and hierarchical feature extraction. The extracted features from these grids represent the unique characteristics of the frames and serve as input for classification. Finally, the frames are categorized into distinct cases (Case 1, Case 2, Case 3, and Case 4) based on predefined criteria such as motion patterns, object detection, or scene analysis, facilitating detailed analysis and understanding of the video content as shown in Fig 12.

## 4 Results and analysis

Our baseline assessment was done through a human-judged approach. A table with four entries was created for: Case A (No Motion), Case B (Camera Motion Only), Case C (Object Motion), and Case D (All Motion). The target users were shown several video scenes and asked to identify the class of motion for each. The responses from the human-judged

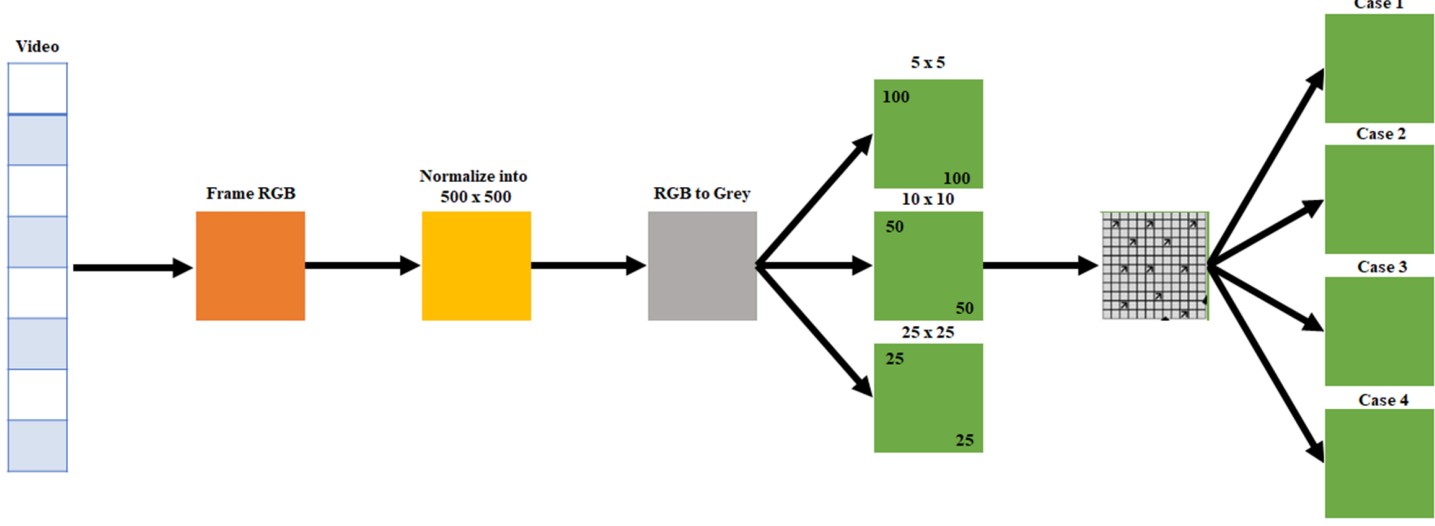

**Fig 12**. Proposed methodology work flow for motion detection and classification.

approach were compared with the algorithm's results, yielding 90% agreement. Based on this comparison, we conclude that our algorithm has attained an accuracy level of 90%.

The motion detection and analysis system processes video frames of horses running along a beach to capture and quantify dynamic elements of the scene. The system employs multiple components to offer diverse perspectives in Figs 13, 14, 15, and 16. (a) the original image represents the unaltered video frame, in Fig 13 showcasing the horses with a backdrop of crashing waves, in Fig 14 showcasing the birds movements in a static background, in Fig 15 showcasing the movement of bird flying bird and its shadow on the ground while other objects are in static position, in Fig 16 both special scenario is addressed in which all motion types are present i.e. camera motion, object motion and background motion, (b) A grayscale image removes color information, simplifying data, retaining only intensity values, which aids in motion detection by reducing computational complexity. (c) The motion-containing part isolates and highlights regions of significant motion, identified through changes in pixel intensity, primarily corresponding to the movement of the horses and waves in Fig 13, movement of birds in Fig 14, movement of a single bird and its shadow on the ground in a complex background in Fig 15, and the movement of camera, object and background in Fig 16, (d) A 3D bar chart of pixel intensities visualizes the grayscale image's brightness distribution, where the height of each bar indicates the number of pixels at a particular intensity. (e) The motion vector plot illustrates the direction and magnitude of detected motion using arrows, with length representing speed and direction indicating movement trajectory. This multi-faceted approach facilitates a comprehensive analysis of motion within the scene. The grayscale transformation simplifies feature extraction, while the motion-containing part identifies areas of interest, enabling the detection of dynamic changes such as the movement of horses and waves in Fig 13, movement of birds in Fig 14, movement of single bird and its shadow in Fig 15, movement of objects, camera and background in Fig 16, The pixel intensity bars provide a statistical overview of brightness distribution, helpful for monitoring environmental lighting variations. The motion vector plot quantitatively represents motion patterns, providing insights into speed and direction and enabling the identification of motion types, such as object motion, background motion, camera motion, and motion involving both background and foreground elements. These features are particularly useful for applications such as video analysis, enabling the tracking of objects and for advancing computer vision research in object tracking and scene understanding.

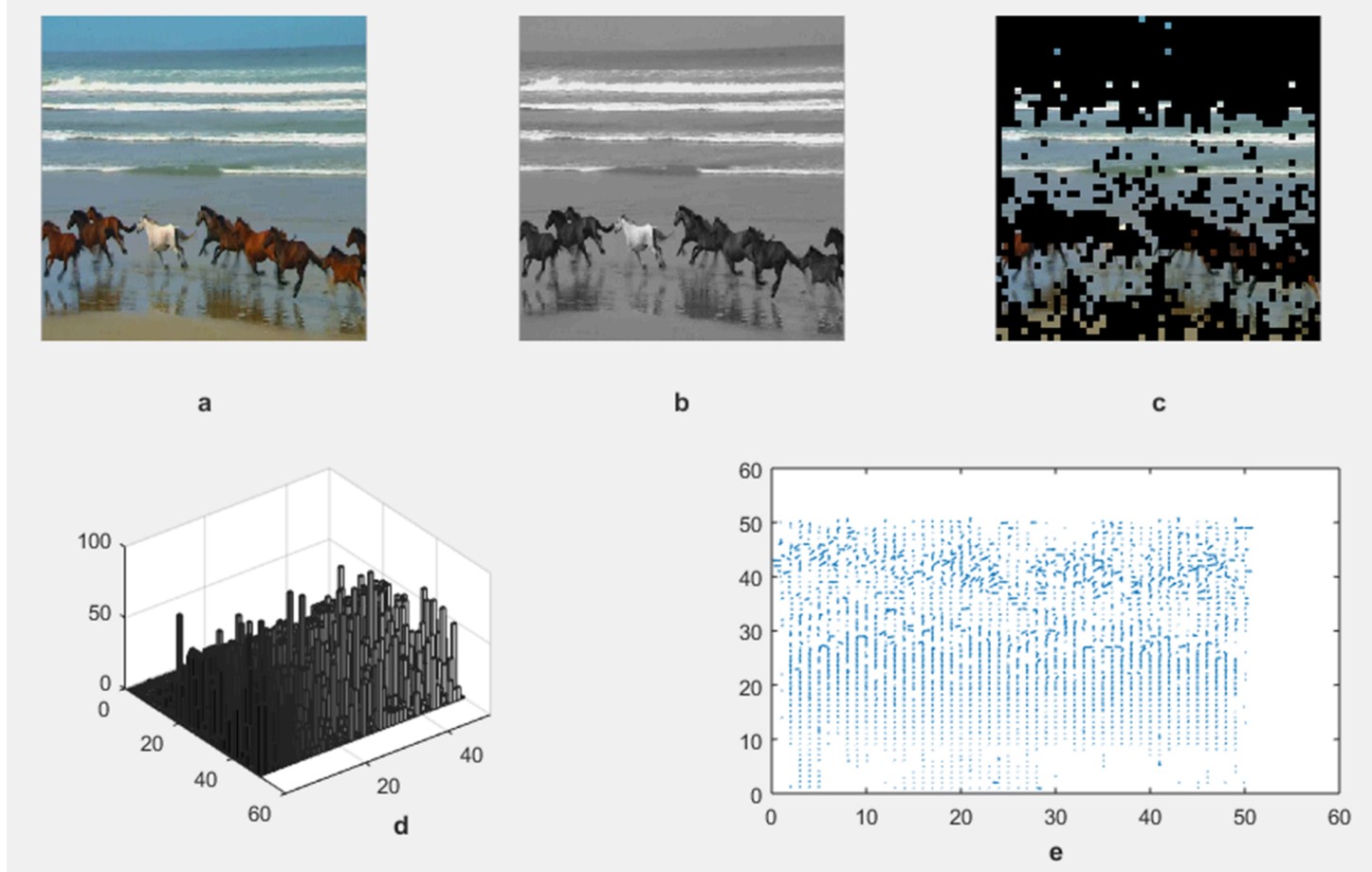

**Fig 13**. **Result of Proposed Methodology (a) The Original Image (b) Grayscale Image (c) Motion Containing Part (d) 3D Bar Chart of Pixel Intensity and (e) Motion Vector Plot.**

## 4.1 Dataset expansion and robustness testing

To test the model's adaptability, we extended the evaluation to include sequences from publicly available datasets, such as SBI2015. From these datasets, we use the hall monitor shown in Fig 17, and custom video as shown in Fig 18. These datasets feature challenges such as variable lighting, dynamic weather, and shadows. Our model retained a performance level similar to the custom dataset used in our initial study. This reinforces the robustness and flexibility of our method across diverse scenarios without requiring retraining.

## 4.2 Experimental results

In the current study, the dataset lacks annotated ground truth labels for motion classes (e.g., object, camera, complex motion), which makes direct computation of these metrics infeasible. To address this limitation, we adopted a human judgment-based validation approach, where motion classification outputs were independently reviewed by multiple observers. Agreement between the algorithm and consensus human classification was used as a proxy for accuracy. While this approach introduces subjectivity, it provides a reasonable basis for preliminary evaluation in the absence of

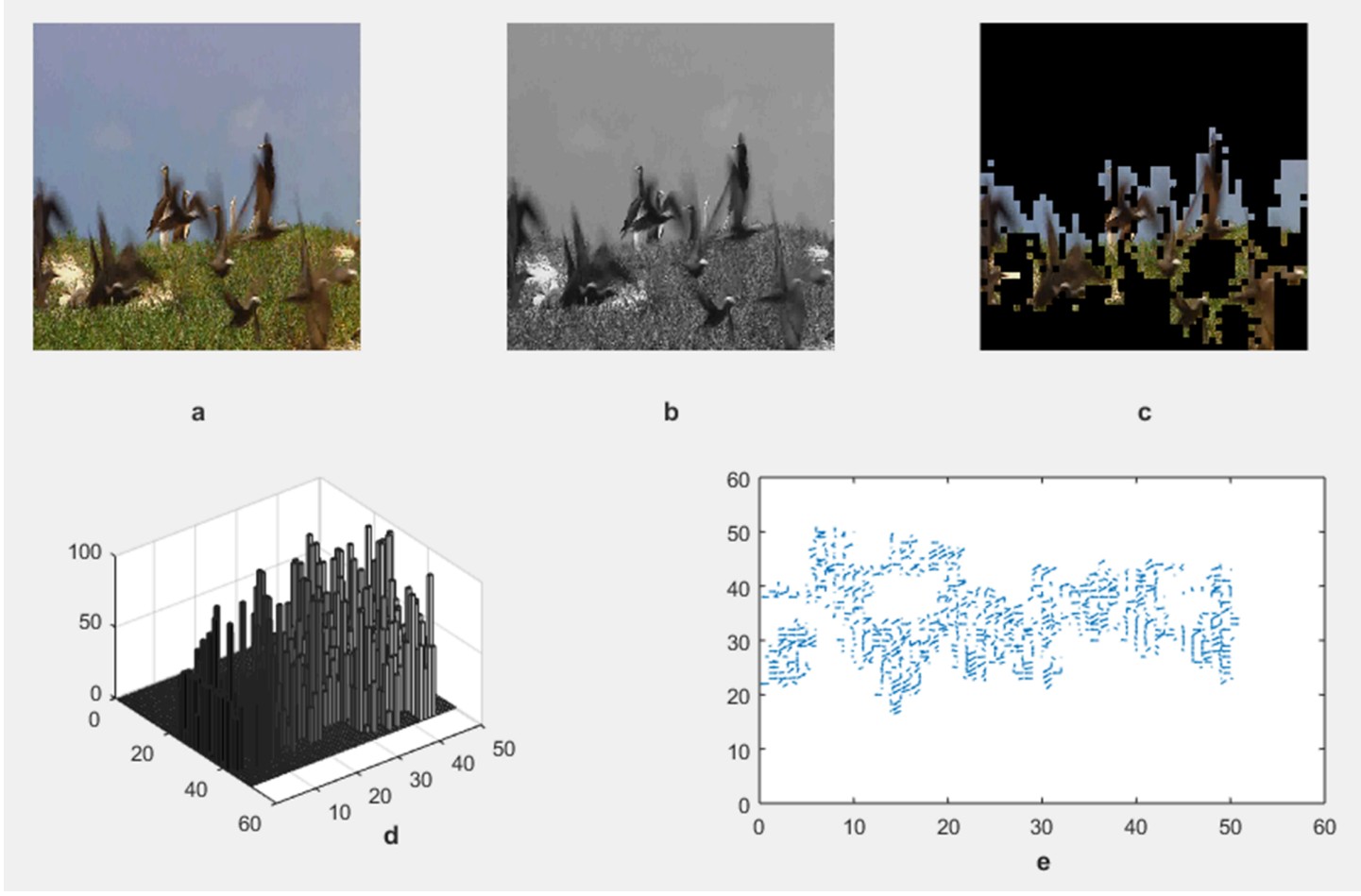

**Fig 14. Result of Proposed Methodology (a) The Original Image (b) Grayscale Image (c) Motion Containing Part (d) 3D Bar Chart of Pixel Intensity and (e) Motion Vector Plot.**

labelled data. Additionally, we have included processing speed metrics as an objective measure of performance. The proposed method processes frames at an average rate of 28 frames per second on a standard Intel i7 CPU (without a GPU), demonstrating its suitability for practical applications. The provided data Table 3 categorizes the motion types in various video files by percentage of total frames. In the video "paris_cif.avi" with 1,065 frames, 0.6% have no motion, none exhibit camera motion, 94.0% show object motion, and 5.33% involve complex motion. "Mixmasterfile.mpg" has 34,389 frames, with 3.4% showing no motion, 20.1% showing camera motion, 41.2% depicting object motion, and 34.4% involving complex motion. "Aeroplans.mp4," with 2,049 frames, has 0.1% with no motion, 45.6% with camera motion, no object motion, and 53.4% complex motion. "Wildlife.avi," containing 902 frames, has 3.0% no motion, 58.5% camera motion, no object motion, and 39.8% complex motion. Finally, "Pool.mp4" with 2,705 frames shows 28.1% no motion, 11.5% camera motion, 48.3% object motion, and 7.9% complex motion.

The Table 4 presents human judgmental results on the types of motion observed across different scenes in five video files, conducted for our research purposes. In "paris_cif.avi" (7 scenes), none of the scenes show no motion or camera motion, 6 scenes involve object motion, and 1 scene has complex motion. "Mixmasterfile.mpg" (85 scenes) contains 3 scenes with no motion, 17 with camera motion, 35 with object motion, and 30 with complex motion. "Aeroplans.mp4"

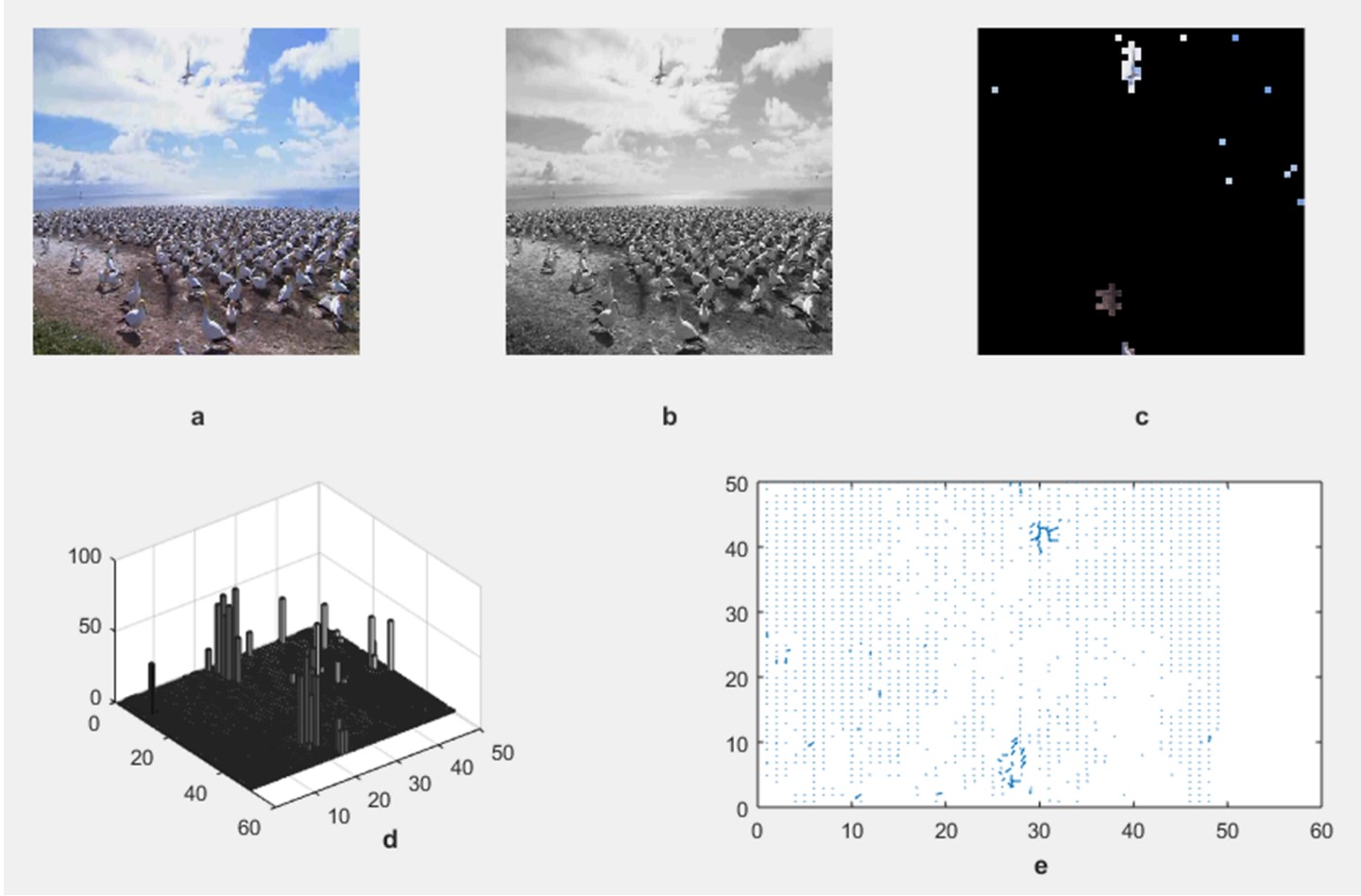

**Fig 15.** **Result of Proposed Methodology (a) The Original Image (b) Grayscale Image (c) Motion Containing Part (d) 3D Bar Chart of Pixel Intensity and (e) Motion Vector Plot.**

(11 scenes) has no scenes with no motion, 5 with camera motion, none with object motion, and 6 with complex motion. "Wildlife.avi" (10 scenes) shows no scenes with no motion, 6 with camera motion, none with object motion, and 4 with complex motion. Finally, "Pool.mp4" (17 scenes) includes 5 scenes with no motion, 2 with camera motion, 9 with object motion, and 1 with complex motion.

The Table 5 compares human judgmental results with the proposed model results for different motion types in "Paris_Cif.avi." Both the human assessment and the proposed model identified no scenes with no motion (Case A) or camera motion (Case B). For object motion (Case C), both methods agreed on 6 scenes. In the case of complex motion (Case D), both the human judgment and the proposed model recognized 1 scene. Overall, the comparison shows complete agreement between the human judgmental results and the proposed model results, with both methods identifying 6 scenes with object motion and 1 scene with complex motion, totaling 7 scenes.

The Table 6 compares human judgmental results with proposed model results for various motion types in "Mixmaster-file.mpg." For scenes with no motion (Case A), the proposed model identified 4 scenes, compared to 3 scenes judged by humans. In scenes with camera motion (Case B), the model found 19 scenes, closely aligning with the 17 identified by

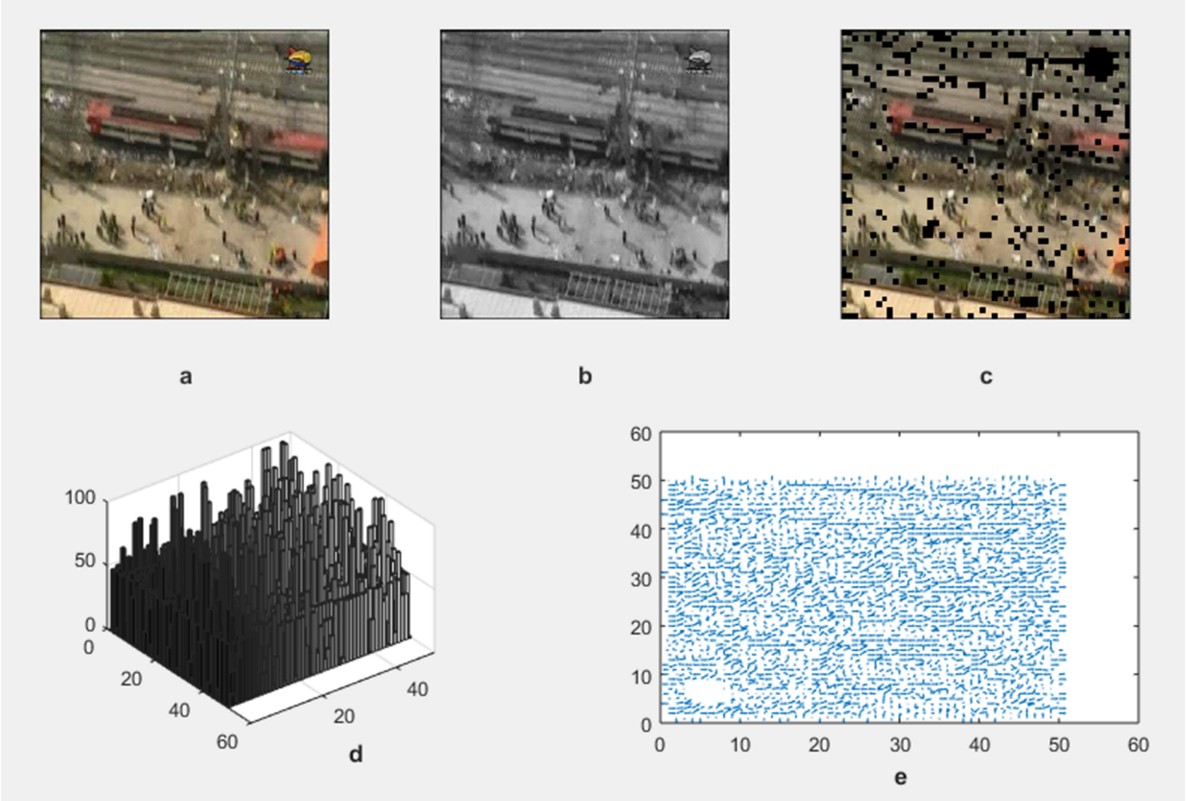

**Fig 16**. **Result of Proposed Methodology (a) The Original Image (b) Grayscale Image (c) Motion Containing Part (d) 3D Bar Chart of Pixel Intensity and (e) Motion Vector Plot.**

humans. For object motion (Case C), the model recognized 34 scenes, while humans identified 35. In the case of complex motion (Case D), the model identified 28 scenes, compared to 30 scenes in human judgment. Overall, the total scene count is consistent at 85 for both methods, with minor variations in classification across different motion types. This comparison indicates a high level of agreement between the human judgment and the proposed model, with slight discrepancies in categorizing specific scenes.

Table 7 compares human judgmental results with proposed model results for different motion types in "Aeroplanes.mp4." Both the human assessment and the proposed model agreed that there were no scenes with no motion (Case A) or object motion (Case C). For camera motion (Case B), both methods identified 5 scenes. In the case of complex motion (Case D), both the human judgment and the proposed model recognized 6 scenes. Overall, the comparison shows complete agreement between the human judgmental results and the proposed model results, with both methods consistently identifying the same number of scenes across all motion categories, totaling 11 scenes.

In Table 8 the comparison between human judgmental results and proposed model results for "Wildlife.avi" shows complete alignment across all motion categories. Both the human assessment and the proposed model agreed that there were no scenes with no motion (Case A) or object motion (Case C). For scenes with camera motion (Case B), both methods identified 6 scenes. Likewise, in scenes categorized as complex motion (Case D), both the human judgment and the proposed model recognized 4 scenes. Overall, there is perfect agreement between the human judgmental results and the proposed model results, with both methods consistently identifying the same number of scenes across all motion categories, totaling 10 scenes.

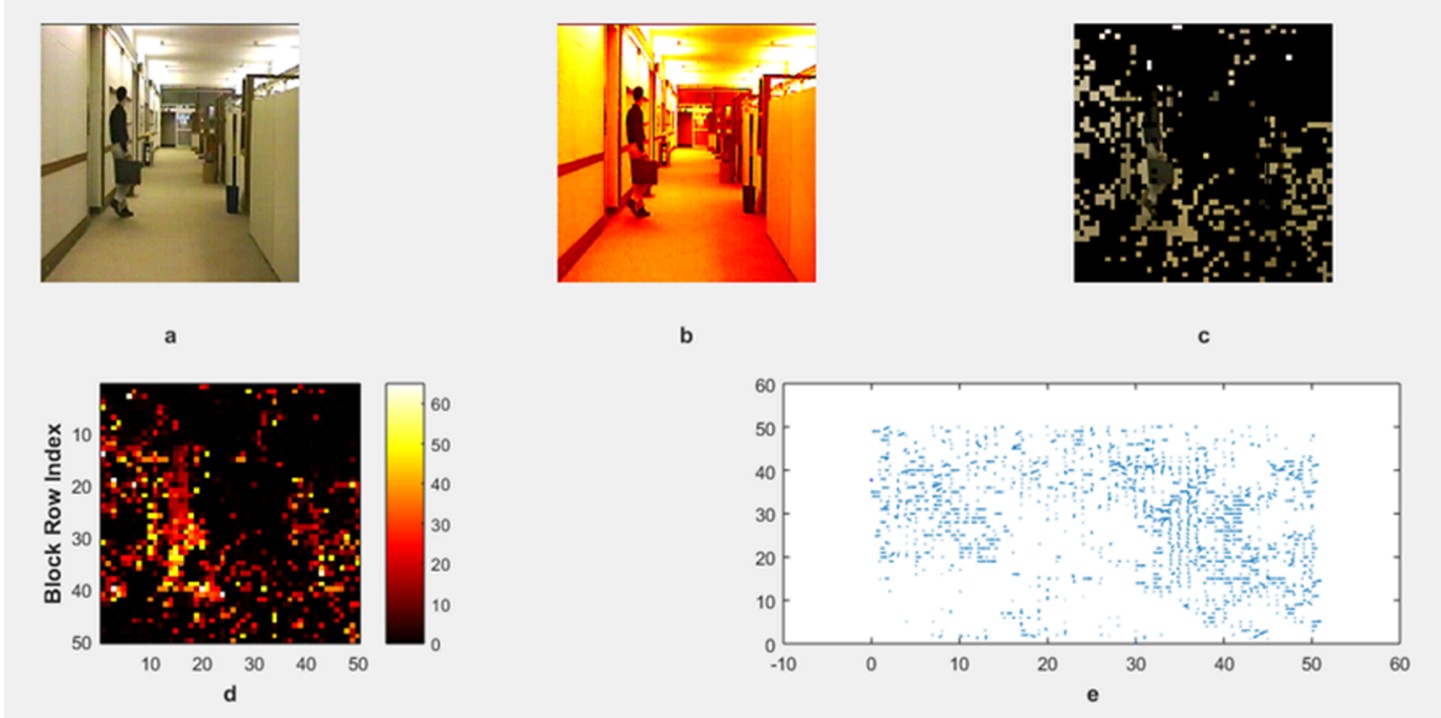

**Fig 17**. **Result of Proposed Methodology (a) The Original Image (b) Grayscale Image (c) Motion Containing Part (d) Heat map (e) Motion Vector Plot.**

In Table 9 the comparison between human judgmental results and proposed model results for "Pool.mp4" reveals a high level of agreement across most motion categories. Both the human assessment and the proposed model identified 5 scenes with no motion (Case A). For scenes with camera motion (Case B), both methods identified 2 scenes. In scenes categorized as object motion (Case C), both the human judgment and the proposed model recognized 9 scenes. Similarly, in scenes characterized as complex motion (Case D), both methods identified 1 scene. Overall, there is substantial agreement between the human judgmental results and the proposed model results, with both methods consistently identifying the same number of scenes across most motion categories, totaling 17 scenes.

### 4.3 Comparison with existing motion detection algorithms

**4.3.1 Computational efficiency.**  Macro block-based motion analysis avoids complex deep learning inference stages. At the same time, it is block-wise only comparison between frames, which significantly leads to time computation reduction and hence applicability in real-time on low-power devices.

**4.3.2 No extensive training data required.**  Macro block methods, unlike deep learning object detection models like YOLO and Faster R-CNN, are self-sufficient and do not call for training on annotated data. Thus, they are data-agnostic and much simpler to deploy in new environments or non-seen conditions.

**4.3.3 Robustness to background clutter and occlusion.**  It analyzes motion block-wise, thus the differentiation between motion of the background and that occurring on the foreground continues, despite cluttered or dynamic scenes. It employs more direct and geometry-aware techniques toward moving region detection, especially when the object is partly occluded.

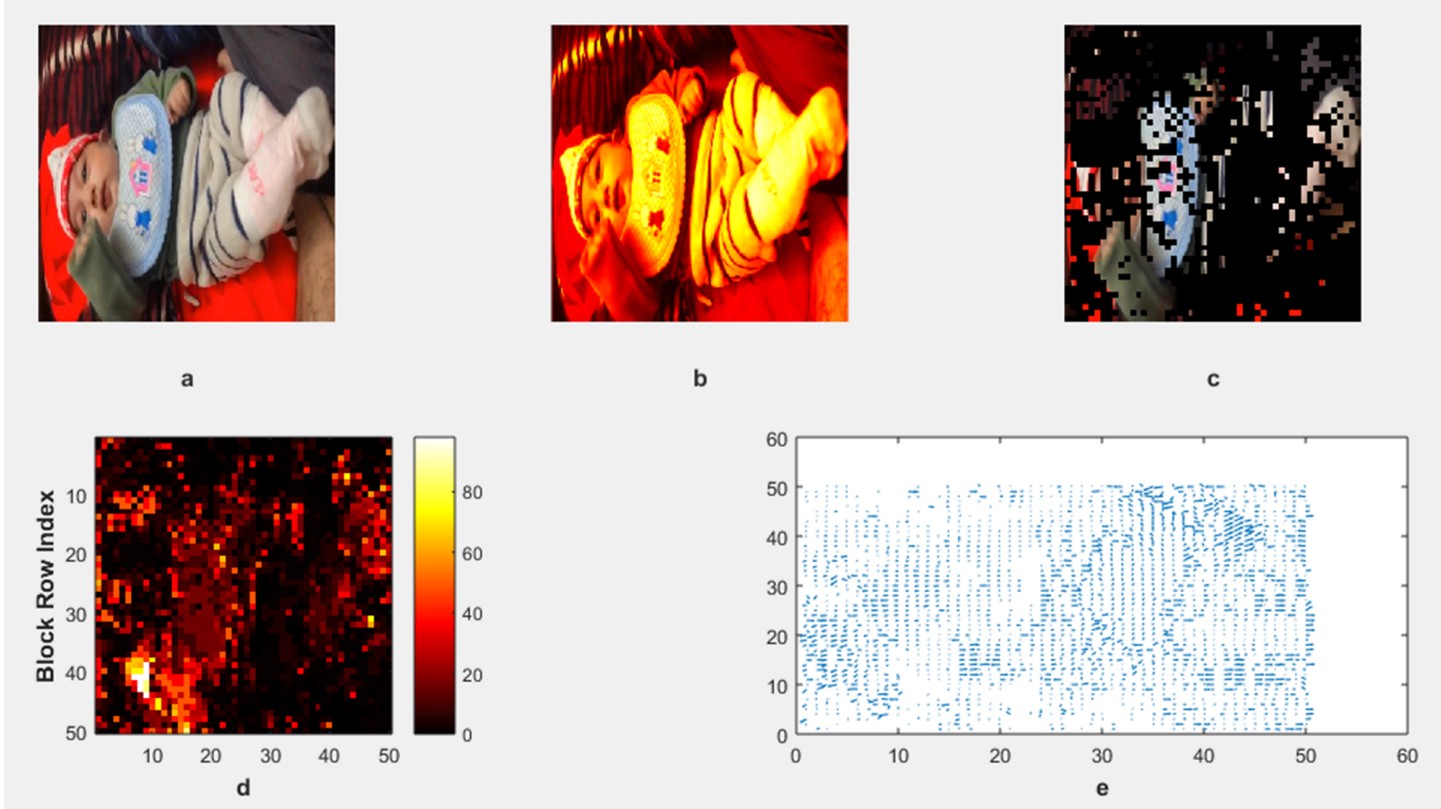

**Fig 18**. **Custom video for more accurate testing.**

**Table 3**. **Summary chart for human judgmental results.**

| Video | Total Frames | CASE A: No Motion (%) | CASE B: Camera Motion (%) | CASE C: Object Motion (%) | CASE D: All/Complex Motion (%) |
|---|---|---|---|---|---|
| paris_cif.avi | 1,065 | 0.6 | 0.0 | 94.0 | 5.33 |
| Mixmasterfile.mpg | 34,389 | 3.4 | 20.1 | 41.2 | 34.4 |
| Aeroplans.mp4 | 2,049 | 0.1 | 45.6 | 0.0 | 53.4 |
| Wildlife.avi | 902 | 3.0 | 58.5 | 0.0 | 39.8 |
| Pool.mp4 | 2,705 | 28.1 | 11.5 | 48.3 | 7.9 |

**Table 4**. **Comparison between proposed VS human experiment result.**

| VIDEOS | Total Scenes | CASE A NO MOTION | CASE B CAMERA MOTION | CASE C OBJECT MOTION | CASE D COMPLEX MOTION |
|---|---|---|---|---|---|
| paris_cif.avi | 7 | 0 | 0 | 6 | 1 |
| Mixmasterfile.mpg | 85 | 3 | 17 | 35 | 30 |
| Aeroplans.mp4 | 11 | 0 | 5 | 0 | 6 |
| Wildlife.avi | 10 | 0 | 6 | 0 | 4 |
| Pool.mp4 | 17 | 5 | 2 | 9 | 1 |

**4.3.4 Adapting to scene variability.** Relative motion and not appearance features is what macro block motion vectors are designed to capture. This means that they would be much less affected by changes in illumination, shadows, or appearance variations-pretty common conditions that severely degrade the performance of visual-based object detectors.

**Table 5**. Confusion Matrix: Human Judgement (Ground Truth) vs. Proposed Method (Predicted).

| Human/Predicted | Case A | Case B | Case C | Case D | Total |
|---|---|---|---|---|---|
| Case A: No Motion | 0 | 0 | 0 | 0 | 0 |
| Case B: Camera Motion | 0 | 0 | 0 | 0 | 0 |
| Case C: Object Motion | 0 | 0 | 6 | 0 | 6 |
| Case D: Complex Motion | 0 | 0 | 0 | 1 | 1 |
| Total | 0 | 0 | 6 | 1 | 7 |

**Table 6**. Comparison between human judgment and proposed method on Mixmasterfile.Mpg.

| Human/Predicted | Case A | Case B | Case C | Case D | Total |
|---|---|---|---|---|---|
| Case A: No Motion | 2 | 0 | 1 | 1 | 4 |
| Case B: Camera Motion | 1 | 15 | 2 | 1 | 19 |
| Case C: Object Motion | 0 | 1 | 31 | 2 | 34 |
| Case D: Complex Motion | 0 | 1 | 1 | 26 | 28 |
| Total | 3 | 17 | 35 | 30 | 85 |

**Table 7**. Comparison between human judgment and proposed method on *Aeroplanes.Mp4*.

| Human/Predicted | Case A | Case B | Case C | Case D | Total |
|---|---|---|---|---|---|
| Case A: No Motion | 0 | 0 | 0 | 0 | 0 |
| Case B: Camera Motion | 0 | 5 | 0 | 0 | 5 |
| Case C: Object Motion | 0 | 0 | 0 | 0 | 0 |
| Case D: Complex Motion | 0 | 0 | 0 | 6 | 6 |
| Total | 0 | 5 | 0 | 6 | 11 |

**Table 8**. Comparison between human judgment and proposed method on *Wildlife.Avi*.

| Human/Predicted | Case A | Case B | Case C | Case D | Total |
|---|---|---|---|---|---|
| Case A: No Motion | 0 | 0 | 0 | 0 | 0 |
| Case B: Camera Motion | 0 | 6 | 0 | 0 | 6 |
| Case C: Object Motion | 0 | 0 | 0 | 0 | 0 |
| Case D: Complex Motion | 0 | 0 | 0 | 4 | 4 |
| Total | 0 | 6 | 0 | 4 | 10 |

**Table 9**. Comparison between human judgment and proposed method on *Pool.Mp4*.

| Human/Predicted | Case A | Case B | Case C | Case D | Total |
|---|---|---|---|---|---|
| Case A: No Motion | 5 | 0 | 0 | 0 | 5 |
| Case B: Camera Motion | 0 | 2 | 0 | 0 | 2 |
| Case C: Object Motion | 0 | 0 | 9 | 0 | 9 |
| Case D: Complex Motion | 0 | 0 | 0 | 1 | 1 |
| Total | 5 | 2 | 9 | 1 | 17 |

## 4.4 Discussion

The adopted methodology of this study is, detection of motion in dynamic backgrounds through a well-structured pre-processing, feature extraction, and motion analysis system. It begins with data collection, in which videos from different sources are gathered to form a diversified dataset essential for robust motion detection and classification. During the pre-processing stage, frame extraction and normalizing were involved, frames were normalized to address lighting and exposure differences, contrast, and exposure differences, ensuring that the algorithm considers the real motion changes rather than brightness and contrast differences as a basis for detecting motion. Subsequently, the frames were resized to a uniform 50×50 resolution and converted to grayscale. This standardization was vital for eliminating the color variations and

emphasizing the intensity difference that augmented the motion detection accuracy. The frames were further segmented into macro blocks of 10 × 10 for high computational efficiency and the ability to capture minor details of motion. Motion vectors for these blocks were predicted based on the difference between the consecutive frames such that displacement and direction of motion can be specified in polar coordinates. Motion analysis entailed classifying cases into four kinds: no motion, camera motion, object motion, and complex motion. This classification became vital to differentiate various cases of motion like immobile background, mobile camera, or a scene containing many moving things. For example, results from the proposed model had human judgment comparisons showing a favorable correlation. A typical case for this was in the video "paris_cif.avi," where similar judgment was made between the model and human evaluators regarding scenes showing object and complex motion. Such remarkable levels of agreement were observed in the other videos with a few differences in classification, thereby confirming the reliability of the model. The proposed model combines block and pixel-based motion evaluation to replace missing areas in the detection of foreground objects. All these methods, along with the combination of those concepts, make it a perfect algorithm for motion detection in a complicated scenario of the video. The model has achieved compliance with human judgment up to 90% thus making its robustness very clear and its usability in real-life where dynamic background is an important case quite obvious. The method and findings of this study contribute a lot to multitudinous areas of computers, vision, image processing, multimedia, motion estimations, object detection and image segmentation in providing reliable means of complex environment motion sensing in video.

## 5 Conclusion and implications

The present study aims at providing a robust concept of detecting and classifying motion in entirely dynamic surroundings, addressing the limitations of the conventional methods in case of simultaneous movements of the foreground and the background. The proposed system makes the use of some block-wise techniques for motion vector analysis to successfully differentiate between three types of motion: object motion, background motion, and camera/frame motion. The adaptive nature of the algorithm ensures reliable detection in real-world settings under variable lighting conditions and in unstructured environments. Consequently, this increases the algorithms' applicability to diverse fields such as surveillance, autonomous navigation, and robotics. Experimental results provide good evidence for a marked increase in accuracy and robustness, thus validating the practical relevance of the proposed approach. This investigation not only advances the field of motion analysis but also paves the way for future developments in dynamic motion detection and classification efforts, thus paving the way for innovations in computer vision systems.In future we will evaluate the method on publicly available datasets with motion ground truth (e.g., CDNet2014) to compute full precision-recall-F1 metrics and to further strengthen the empirical validation.

## Author contributions

**Data curation:** Sameed Ur Rehman, Irshad Ullah.

**Formal analysis:** Insaf Ullah.

**Resources:** Altaf Hussain.

**Software:** Shuguang Li.

**Supervision:** Ahmad Ali AlZubi.

**Validation:** Wajahat Akbar.

**Writing – original draft:** Sameed Ur Rehman.

**Writing – review & editing:** Tariq Hussain.

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
