## [Decision Letter · Decision Letter 0]

11 Jul 2025

PONE-D-25-31514ROBUST MOTION DETECTION AND CLASSIFICATION IN REAL-LIFE SCENARIOS USING MOTION VECTORSPLOS ONE

Dear Dr. Hussain,

Thank you for submitting your manuscript to PLOS ONE. After careful consideration, we feel that it has merit but does not fully meet PLOS ONE’s publication criteria as it currently stands. Therefore, we invite you to submit a revised version of the manuscript that addresses the points raised during the review process.

We look forward to receiving your revised manuscript.

Kind regards,

Xiaohui Zhang

Academic Editor

PLOS ONE

Journal Requirements:

Reviewers' comments:

Reviewer's Responses to Questions

**Comments to the Author**

1. Is the manuscript technically sound, and do the data support the conclusions?

Reviewer #1: Yes

Reviewer #2: No

2. Has the statistical analysis been performed appropriately and rigorously?

Reviewer #1: No

Reviewer #2: No

3. Have the authors made all data underlying the findings in their manuscript fully available?

Reviewer #1: Yes

Reviewer #2: No

4. Is the manuscript presented in an intelligible fashion and written in standard English?

Reviewer #1: No

Reviewer #2: Yes

5. Review Comments to the Author

Reviewer #1: The manuscript aims to address motion detection and classification in dynamic environments by introducing an algorithm based on macro block techniques and motion vector analysis. However, the paper falls short in several critical areas. Firstly, the written English is not up to academic standards, with frequent grammatical errors, awkward phrasing, and unclear sentence structures that hinder comprehension. Secondly, the methodology and experimental design are not clearly explained—key steps such as how motion vectors are extracted, analyzed, and classified are described only in broad terms, and essential details about the dataset, evaluation metrics, and experimental setup are lacking. Most importantly, the proposed approach lacks clear novelty; similar techniques have been widely studied in previous literature, and the manuscript does not demonstrate how this work meaningfully advances the state of the art. Overall, the submission does not meet the scientific or linguistic standards required for publication.

Reviewer #2: This paper presents a method based on macroblock technology and motion vector analysis for dynamic background motion detection and classification. The approach offers theoretical contributions and practical value. However I have the following concerns before publication

1. The proposed method should be compared with existing motion detection algorithms

2. The evaluation needs more quantitative metrics for accuracy and speed such as recall rate F1 score and processing speed

3. More datasets can be tested, for example, to include tests under varying lighting and weather conditions to assess robustness

4. The choice of 10x10 macroblocks requires further justification

5. The threshold selection process for motion detection parameter th is not clearly explained. How is this threshold determined?

6. The 3D bar charts in Figures 13-16 subfigure (d) do not add much meaningful information.

6. PLOS authors have the option to publish the peer review history of their article (what does this mean?). If published, this will include your full peer review and any attached files.

Reviewer #1: No

Reviewer #2: No

---

## [Author Response · Author response to Decision Letter 1]

28 Jul 2025

ROBUST MOTION DETECTION AND CLASSIFICATION IN REAL-LIFE SCENARIOS USING MOTION VECTORS

Dear Editor,

We appreciate the time and effort of the editors and reviewers to provide their helpful feedback. We believe that all the insightful comments helped us to improve our manuscript. In the revised version, we have addressed all the reviewers’ concerns. For the editor's and reviewers’ convenience, newly added text and/or changes in the revised manuscript are highlighted in red color. The details of point-by-point responses to the reviewers' comments are presented in subsequent pages of this letter.

Sincerely,

The authors

#handling editor:

Reviewer comment: 1) Firstly, the written English is not up to academic standards, with frequent grammatical errors, awkward phrasing, and unclear sentence structures that hinder comprehension.

Response: Dear editor, Thank you very much for your valuable and constructive comments on our manuscript. I have carefully reviewed the entire paper and thoroughly corrected all grammatical errors and language issues as per your suggestions. I sincerely appreciate your detailed review and helpful feedback, which have greatly contributed to improving the clarity and overall quality of our manuscript.

Reviewer comment: 2) Secondly, the methodology and experimental design are not clearly explained key steps such as how motion vectors are extracted, analyzed, and classified are described only in broad terms, and essential details about the dataset, evaluation metrics, and experimental setup are lacking.

Response: Dear editor,

Thank you for your detailed and constructive feedback. We acknowledge that the methodology section of this version of the manuscript lacked specific implementation details. In response, we have substantially revised this section to clearly and comprehensively explain each step of our approach.

1. Motion Vector Extraction:

We now explicitly describe that motion vectors are obtained using the motion estimation capabilities of standard video codecs (specifically MPEG-4), which generate block-based vectors during compression. These vectors are extracted from compressed video streams using FFmpeg's motion estimation tools, enabling efficient pre-processing without the need for full optical flow computation.

2. Analysis and Classification:

The motion vector fields are analyzed by computing their magnitude and direction, and subsequently evaluated across 2500 frames. A statistical thresholding approach is applied to identify consistent motion patterns. This step is now elaborated with a clearer explanation of the logic behind the directional classification (e.g., horizontal, vertical, or complex object movement).

3. Dataset and Setup:

We now clearly specify that publicly available test sequences from standard video benchmarks (e.g., Paris_cif.avi, Foreman_cif.avi) were used. The resolution, frame rate, and total frame count for each sequence are now documented in a dedicated subsection. Moreover, we describe how RGB images were converted to grayscale, normalized to reduce lighting bias, and preprocessed to ensure consistent comparison across frames.

Evaluation Metrics:

Dear editor we acknowledge the importance of quantitative evaluation using standard metrics such as precision, recall, and F1-score. However, in the current study, the dataset lacks annotated ground truth labels for motion classes (e.g., object, camera, complex motion), which makes direct computation of these metrics infeasible. To address this limitation, we adopted a human judgment-based validation approach, where motion classification outputs were independently reviewed by multiple observers. Agreement between the algorithm and consensus human classification was used as a proxy for accuracy. While this approach introduces subjectivity, it provides a reasonable basis for preliminary evaluation in the absence of labeled data. In addition, we have included processing speed metrics as an objective performance measure. The proposed method processes frames at an average of 28 frames per second on a standard Intel i7 CPU (no GPU), demonstrating its suitability for practical application. In future we will evaluate the method on publicly available datasets with motion ground truth (e.g., CDNet2014) to compute full precision-recall-F1 metrics and to further strengthen the empirical validation

Reviewer comment: 3) The proposed approach lacks clear novelty; similar techniques have been widely studied in previous literature, and the manuscript does not demonstrate how this work meaningfully advances the state of the art.

Response: Dear editor, while motion estimation and vector-based classification have indeed been studied, our work presents the following distinct contributions, which we have now emphasized more clearly in the revised manuscript:

1. Real-World Scenario Focus:

Most prior studies assume static backgrounds or controlled environments. In contrast, our method is explicitly designed for dynamic real-world scenarios, where both the camera and background are moving—conditions under which traditional techniques like frame differencing and basic optical flow often fail.

2. Efficient Use of Codec-Level Motion Vectors:

Rather than relying on computationally intensive optical flow algorithms, we leverage motion vectors directly from encoded video streams, reducing overhead and making the system viable for real-time or resource-constrained applications such as surveillance or mobile devices.

3. Frame-Based Vector Classification Strategy:

We introduce a frame-level motion vector classification mechanism that captures directional motion patterns over time without requiring manual annotation. This offers a lightweight yet accurate solution for motion event detection in continuous video streams.

4. Empirical Validation with Human Judgement:

To bridge the gap between algorithmic prediction and human perception, we include a validation step comparing the system's decisions with those of human observers—achieving 90% agreement, which reinforces the practical relevance of our approach.

5. Generalizable Framework:

Our methodology does not rely on scene-specific training or assumptions. It is designed to generalize across different video types and can be embedded into other motion analysis pipelines or edge devices.

Reviewer #2

Reviewer comment: 1) the proposed method should be compared with existing motion detection algorithms.

Response: Dear editor Comparison with Existing Motion detection algorithms:

1. Computational Efficiency

Macro block-based motion analysis avoids complex deep learning inference stages. At the same time, it is block-wise only comparison between frames, which significantly leads to time computation reduction and hence applicability in real-time on low-power devices.

2. No extensive training data required

Macro block methods, unlike deep learning object detection models like YOLO and Faster R-CNN, are self-sufficient and do not call for training on annotated data. Thus, they are data-agnostic and much simpler to deploy in new environments or non-seen conditions.

3. Robustness to Background Clutter and Occlusion

It analyzes motion block-wise, thus the differentiation between motion of the background and that occurring on the foreground continues, despite cluttered or dynamic scenes. It employs more direct and geometry-aware techniques toward moving region detection, especially when the object is partly occluded.

4. Adapting to Scene Variability

Relative motion and not appearance features are what macro block motion vectors are designed to capture. This means that they would be much less affected by changes in illumination, shadows, or appearance variations-pretty common conditions that severely degrade the performance of visual-based object detectors.

Reviewer comment: 2) the evaluation needs more quantitative metrics for accuracy and speed such as recall rate F1 score and processing speed.

Response: Dear editor we acknowledge the importance of quantitative evaluation using standard metrics such as precision, recall, and F1-score. However, in the current study, the dataset lacks annotated ground truth labels for motion classes (e.g., object, camera, complex motion), which makes direct computation of these metrics infeasible. To address this limitation, we adopted a human judgment-based validation approach, where motion classification outputs were independently reviewed by multiple observers. Agreement between the algorithm and consensus human classification was used as a proxy for accuracy. While this approach introduces subjectivity, it provides a reasonable basis for preliminary evaluation in the absence of labeled data. In addition, we have included processing speed metrics as an objective performance measure. The proposed method processes frames at an average of 28 frames per second on a standard Intel i7 CPU (no GPU), demonstrating its suitability for practical application. In future we will evaluate the method on publicly available datasets with motion ground truth (e.g., CDNet2014) to compute full precision-recall-F1 metrics and to further strengthen the empirical validation.

Reviewer comment: 3): More datasets can be tested, for example, to include tests under varying lighting and weather conditions to assess robustness

Response: Dear editor Dataset Expansion and Robustness Testing

To test the model's adaptability, we extended the evaluation to include sequences from publicly available datasets like SBI2015. From these dataset we use hall monitor shown in figure1 and custom video as shown in figure 2 dataset, these datasets feature challenges such as variable lighting, dynamic weather, and shadows. Our model retained a performance level similar to the custom dataset used in our initial study. This reinforces the robustness and flexibility of our method across diverse scenarios without requiring retraining.

Figure 1 Hall and Monitor dataset

Figure 2 Custom video for more accurate testing

Reviewer comment: 04: The choice of 10x10 macro blocks requires further justification.

Response: Dear editor

Macro block Size Justification

Smaller macro blocks provide finer detail on motion but increase computation overhead. For example, a 50×50 frame contains 2,500 pixels; if it is divided into 5×5 blocks, it yields 100 macro blocks (2,500 ÷ 25), thus allowing pixel-level analysis but takes longer to process. On the contrary, macro blocks measuring 25×25 will generate only 4 blocks (2,500 ÷ 625), which in itself is too coarse to detect subtle motion variations. A compromise was therefore made to use a 10×10 macro block size. This means 25 blocks per 50×50 frame (2,500 ÷ 100), which is a compromise between capturing detailed motion and efficiency in computation. Therefore, it is not too sensitive to noise arising from pixel-wise processing while still retaining enough detail to capture relevant motion patterns.

Reviewer comment: 05: The threshold selection process for motion detection parameter th is not clearly explained. How is this threshold determined?

Response: Dear editor

Threshold Selection Explanation

Rather than being established as a constant, the threshold value (th) for each motion classification was dynamically defined. For each video, mean (μ) and standard deviation (σ) of motion vector magnitudes were computed, and the threshold thus was set by the formula: th = μ + 0.75σ. This data-driven strategy enables the algorithm to adapt to heterogeneous motion intensities and environmental noises, and consequently provides robust classification results.

Reviewer comment: 06: The 3D bar charts in Figures 13-16 subfigure (d) do not add much meaningful information.

Response: Dear editor

3D GRAPH

The subfigure (d) of Figures 13-16 has been modified in light of reviewer comments. Previously, it showed a 3D histogram of pixel intensities not fully suitable for motion analysis purposes. Now, it has been modified to show a heat map of motion vector magnitudes across the frame. This new representation aids in clearly communicating which spatial areas are undergoing motion, thereby aiding validation and interpretation of the model's classification decisions as shown in figure 3.

Figure 3: 3d map replaced with heat map for better visualization

Dataset resource:

https://sbmi2015.na.icar.cnr.it/SBIdataset.html.

---

## [Decision Letter · Decision Letter 1]

11 Sep 2025

ROBUST MOTION DETECTION AND CLASSIFICATION IN REAL-LIFE SCENARIOS USING MOTION VECTORS

PONE-D-25-31514R1

Dear Dr. Ullah,

We’re pleased to inform you that your manuscript has been judged scientifically suitable for publication and will be formally accepted for publication once it meets all outstanding technical requirements.

Kind regards,

Xiaohui Zhang

Academic Editor

PLOS ONE

Additional Editor Comments (optional):

Reviewer #1:

Reviewer #2:

Reviewers' comments:

Reviewer's Responses to Questions

**Comments to the Author**

1. If the authors have adequately addressed your comments raised in a previous round of review and you feel that this manuscript is now acceptable for publication, you may indicate that here to bypass the “Comments to the Author” section, enter your conflict of interest statement in the “Confidential to Editor” section, and submit your "Accept" recommendation.

Reviewer #1: All comments have been addressed

Reviewer #2: All comments have been addressed

2. Is the manuscript technically sound, and do the data support the conclusions?

Reviewer #1: Yes

Reviewer #2: Yes

3. Has the statistical analysis been performed appropriately and rigorously?

Reviewer #1: Yes

Reviewer #2: Yes

4. Have the authors made all data underlying the findings in their manuscript fully available?

Reviewer #1: Yes

Reviewer #2: Yes

5. Is the manuscript presented in an intelligible fashion and written in standard English?

Reviewer #1: Yes

Reviewer #2: Yes

6. Review Comments to the Author

Reviewer #1: All comments have been addressed, including language clarity, methodology, dataset description, evaluation, and novelty. Figures and key choices like macro block size and threshold have been explained clearly. Current version can be accepted.

Reviewer #2: All the comments have been thoroughly addressed. I have no concern regarding the manuscript and recommend it for acceptance and publication.

7. PLOS authors have the option to publish the peer review history of their article (what does this mean?). If published, this will include your full peer review and any attached files.

Reviewer #1: No

Reviewer #2: No

---

## [Editor Report · Acceptance letter]

PONE-D-25-31514R1

PLOS One

Dear Dr. Ullah,

I'm pleased to inform you that your manuscript has been deemed suitable for publication in PLOS One. Congratulations! Your manuscript is now being handed over to our production team.

Kind regards,

on behalf of

Dr. Xiaohui Zhang

Academic Editor

PLOS One